# Revealing the Diversity of Thin Filamentous Cyanobacteria, with the Discovery of a Novel Species, *Pegethrix qiandaoensis* sp. nov. (Oculatellaceae, Oculatellales), in a Freshwater Lake in China

Kaihui Gao [1,†], Yao Cheng [2,†], Rouzhen Geng [3], Peng Xiao [1,4], He Zhang [1,4], Zhixu Wu [5], Fangfang Cai [6,*] and Renhui Li [1,4,*]

1 Zhejiang Provincial Key Laboratory for Subtropical Water Environment and Marine Biological Resources Protection, Wenzhou University, Wenzhou 325035, China; gkh17735571690@163.com (K.G.); pxiao@wzu.edu.cn (P.X.); zhanghe1983004@sina.com (H.Z.)
2 College of Life Sciences and Technology, Harbin Normal University, Harbin 150025, China; wzuchengyao@163.com
3 Research Center for Monitoring and Environmental Sciences, Taihu Basin & East China Sea Ecological Environment Supervision and Administration Authority, Ministry of Ecology and Environment of the People's Republic of China, Shanghai 200125, China; geruozh@163.com
4 National and Local Joint Engineering Research Center of Ecological Treatment Technology for Urban Water Pollution, Wenzhou University, Wenzhou 325035, China
5 Hangzhou Bureau of Ecology and Environment Chun'an Branch, Hangzhou 311700, China; caepb@126.com
6 Hubei Key Laboratory of Animal Nutrition and Feed Science, Wuhan Polytechnic University, Wuhan 430023, China
* Correspondence: fangfangcai@whpu.edu.cn (F.C.); renhui.li@wzu.edu.cn (R.L.)
† These authors contributed equally to this work.

**Abstract:** During the study of diversity in filamentous cyanobacteria in China, two strains (WZU0719 and WZU0723) with the form of thin filaments were isolated from the surface of Qiandao Lake, a large freshwater lake in Zhejiang Province, China. A comprehensive analysis was conducted, incorporating morphological, ecological, and molecular data. The morphological examination provided an initial identification as a *Leptolyngbya*-like cyanobacterium. Genetic characterization was also performed by amplifying the 16S rRNA gene and the 16S-23S rRNA internal transcribed spacer (ITS) region. The phylogenetic grouping based on the 16S rRNA gene demonstrates that the examined strain is unequivocally assigned to the *Pegethrix* genus. However, it possesses distinct phylogenetic divergence from the six described *Pegethrix* species. Additionally, discrepancies in habitat further differentiate it from other members of this genus. Employing the polyphasic approach, we present a comprehensive account of the newly discovered taxa: *Pegethrix qiandaoensis* sp. nov. The novel taxonomic finding in this research significantly contributes to enhancing the comprehension of *Pegethrix* diversity across various habitats.

**Keywords:** new species; 16S rRNA gene; 16S–23S ITS; polyphasic approach

## 1. Introduction

The wide distribution of cyanobacteria, with the nature of microorganisms, makes them an integral component of various ecosystems, including soil and aquatic environments [1,2]. In taxonomic history, cyanobacteria had long been classified based on their morphological characteristics but recent advancements in studying their molecular phylogenies have led to significant revisions [3–5]. Many of the morphological characteristics that were traditionally used as indicators of phylogeny have been proven to be variable. Additionally, phylogenetic reconstructions demonstrated that a large number of genera previously regarded to have consistent morphological traits were polyphyletic [6,7]. These polyphyletic genera, sometimes referred to as cryptogenera, may have resulted from multiple lineages of cyanobacteria undergoing frequent evolutionary convergence [8–10].

The 16S rRNA gene is commonly used as a molecular marker for identifying prokaryotes [11,12], especially in taxonomic studies to determine their assignment at genus level. Moreover, the practical use of secondary structures in the 16S–23S rRNA internal transcribed spacer (ITS) is considered advantageous for discovering new species of cyanobacteria [13,14]. After several major revisions, the current cyanobacterial classification has changed greatly [15,16], including new orders and families proposed to satisfy the requirement of monophyly at all taxonomic ranks. The newly proposed family Oculatellaceae has been demonstrated to make outstanding contributions to further clarifying the relationship between Leptolyngbyales and Pseudanabaenales [17]. Oculatellaceae, primarily consisting of diverse subaerophytic taxa found in wet rocks and soils, was phylogenetically grouped as a sister clade to Pseudanabaenales, with high support [17].

Among the recently described aerophytic genera, *Pegethrix* stands out with its six characterized species. These species have been thoroughly examined in terms of morphology, genetics, and ecology. Interestingly, most of these species could form nodules in their filaments under low light conditions. Furthermore, a species recently discovered in China has been found to exhibit an additional capability of inducing chlorophyll f synthesis under far-red light (FRL) [18]. At present, all species in the genus come from wet rocks and soil crusts near streams [17–19].

In this study, we isolated two thin filamentous cyanobacterial strains (WZU0719 and WZU0723) from the surface of a large freshwater lake in Zhejiang Province, China. We used a polyphasic approach, including molecular characteristics, morphology, and ecological data, to evaluate the taxonomic status of the strains. Initial morphological examination temporarily identified the strains as a *Leptolyngbya*-related morphotype. Molecular analyses, however, determined their position within the genus *Pegethrix* and further supported them as a new species. The main objectives of this study were to clarify the phylogenetic position of the two strains. In addition, the novel species possesses a distinctive habitat, setting it apart from other species within the genus. To differentiate it from other *Pegethrix* species and establish it as a new species, we have named it *Pegethrix qiandaoensis* sp. nov. based on the nomenclature of Algae, Fungi, and Plants. This study presents a taxonomic description of the newly discovered species.

## 2. Materials and Methods

### 2.1. Strain Collection and Cultivation

In June 2020, samples were collected from the surface of Qiandao Lake in Zhejiang Province, China (118°34′ E, 29°22′ N). Samples were transported back to the laboratory and placed in a sampling refrigerator at 4 °C to maintain freshness. Clusters of filaments were separated within the field of view of a dissecting microscope (Carl Zeiss STEMI 508, Oberkochen, Germany) at 40× magnification using a laboratory-made Pasteur pipette. The isolated cell clusters were then placed in 24-well plates containing sterile modified liquid medium BG-11 [20]. After three to four weeks, pure culture was inspected for contaminants and transferred to a screw-cap tube containing 10 mL of the same medium to obtain a unialgal culture. Cultures were exposed to a 12 h light/12 h dark cycle at 25 °C and a photon flux density of 35 μmol·m$^{-2}$·s$^{-1}$. These cultures were deposited at the Algae Culture Collection Center, College of Life and Environmental Science, Wenzhou University, China. The accession numbers of these strains are WZU0719 and WZU0723, respectively.

### 2.2. Morphological Characterization

Live cultures were detected using a LEICA DM2000 LED microscope (LEICA, Wetzlar, Germany) and photomicrographs were taken using a LEICA DMC 5400 photomicrography system (digital camera attached to the microscope) (LEICA, Wetzlar, Germany). Images were analyzed using Leica Application Suite X 3.7.4. Cell size was measured from digital images obtained at 1000× magnification and the average vegetative cell width of more than 60 filaments was calculated.

### 2.3. DNA Extraction, PCR Amplification, and Sequencing

To avoid contamination by other bacteria, the cultured cells were centrifuged and washed three times with sterile phosphorus-free modified liquid medium BG-11. Total genomic DNA of these strains was extracted using a modified cetyltrimethylammonium bromide (CTAB) method [21]. Primers used for 16S-23S rRNA amplification include PA (5′-AGA GTT TGATCC TGG CTC AG-3′) and B23s (5′-CTT CGC CTC TGT GTG CCT AGG T-3′) [22,23]. The 16S rRNA PCR reaction volume was 50 μL, including 2 μL template DNA and 1 μL (10 μmol/L) of each primer, 22 μL sterile water, and 25 μL 2×Taq mix (Dye Plus) (Vazyme Biotech Co. Ltd., Nanjing, China). PCR amplification was performed using a SimpliAmp™ thermal cycler (San Francisco, CA, USA). The PCR profile consisted of an initial denaturation at 95 °C for 5 min, followed by 30 cycles of 95 °C for 30 s, 55 °C for 30 s, and 72 °C for 2 min, and a final extension step of 10 min at 72 °C. PCR products were purified using TIANgel Midi (Tiangen Biotechnology Co., Ltd., Beijing, China) purification kit. The purified PCR products were then cloned using the versatile simple vector kit pClone007 (Beijing Tsingke Biotechnology Co., Ltd., Beijing, China). The insert containing the plasmid in *E. coli* Trelidf™ 5α was replicated using chemically competent cells and incubated at 37 °C for 12–14 h. The clones that contained the target fragment were sequenced bidirectionally using standard plasmid primers M13F (5′-TGT AAA ACG ACG GCC AGT-3′) and M13R (5′-CAG GAA ACA GCT ATG ACC-3′), by Wuhan Tianyi Huayu Gene Technology Co., Ltd. (located in Wuhan, China).

### 2.4. Phylogenetic Analysis

The online tool Nucleotide BLAST was used to compare the 16S-23S rRNA gene sequences obtained in this study. Highly similar 16S-23S rRNA gene sequences were downloaded from the NCBI GenBank database for phylogenetic analysis. Downloaded and captured sequences were aligned using MAFFT v7.463 and trimmed using BioEdit v7.0.9 [24,25]. The phylogenetic tree of 16S rRNA gene outgroup was established using the sequence of *Gloeobacter violaceus* PCC7421. After multiple rounds of sequence comparison, a data matrix containing 1162 sequences and 79 nucleotide sites of the 16S rRNA gene was obtained. The best alternative model for the phylogenetic analysis was selected using the ModelFinder program in the molecular phylogenetic analysis platform PhyloSuite v1.2.2 [26]. Bayesian inference (BI) and maximum-likelihood (ML) analyses were performed based on the Akaike information criterion (AIC). For the 16S rRNA gene matrix, the nucleic acid substitution model (GTR + R4 + F) was chosen for optimal ML analysis, and the recommended fitting model (GTR + I + G + F) was used for BI analysis. The *Pseudanabaena foetida* (LC016779.1) sequence was used as the outgroup to construct the ITS gene phylogenetic tree. After multiple rounds of sequence comparison, a data matrix containing 1080 sequences and 48 nucleotide sites of the ITS gene was obtained. ITS gene matrix analysis used the nucleic acid substitution model (TVM + R5 + F) of maximum-likelihood analysis (ML) and the adaptation model (GTR + G + F) proposed by Bayesian analysis (BI) for phylogenetic analysis. The specific operating parameters of the surrogate model were estimated using IQ-TREE v2.1.3 and Mr-Bayes v3.2.7, respectively [27,28]. For ML analysis, bootstrapping was performed with 1000 pseudo-replicates on default options for relative support. The BI analysis consisted of two parallel runs of 10 million generations, with sampling every 100 generations. The first 25% of sampled data were discarded as burn-in. Neighbor-joining (NJ) analysis was performed using a two-parameter Kimura model with 1000 bootstrap replicates created in MEGA 11. To estimate the genetic distance of the 16S rRNA gene sequence similarity matrix, MEGA 11 used the Kimura 2-parameter model to calculate the p distance of the pairwise deletion gaps. Phylogenetic trees were visualized in Fig Tree v1.4.3, edited in Tree View 1.6.6 [29,30], and adjusted in Adobe Illustrator 2022 software. The sequences of 16S-23S rRNA gene obtained in this study were deposited in the GenBank database with accession numbers OR815386 and PP084669.

### 2.5. Morphological Ultrastructural Characterization

Samples were set with 2.5% glutaraldehyde in 0.1 M phosphate buffer with a pH of 7.2 and a temperature of 4 °C for a duration of three days to stabilize their filaments. Once completed, they underwent three rinses in 0.1 M phosphate buffer. Then, they were subjected to a post-fixation process involving 1% osmium tetroxide for a period of 2 h. Once fixed, the materials went through a dehydration process utilizing an ethanol gradient that progressed from 30% to 50%, then to 70%, 90%, and ultimately 100%. Lastly, the specimens were integrated into Spurr's resin as per the procedure outlined by Spurr in 1969. To facilitate transmission electron microscope (TEM) examination, the samples were sectioned into 80 nm sections using a Leica UC6 ultramicrotome. These sections were subsequently stained with a combination of 2% uranyl acetate and lead citrate. Microscopic observation of the specimen was conducted using a HT7700 transmission electron microscope, operated at 80 kV on a Hitachi TEM system control from Japan.

### 2.6. Analyses of 16S–23S Internal Transcribed Spacer (ITS)

The tRNA gene sequences were detected using the tRNAscan SE 2.0 web server [31] and compared with related taxa by analyzing the 16S-23S rRNA ITS sequences of cyanobacterial strains. To simulate the secondary structure of conserved regions, namely D1-D1′, Box-B, and V3 helices, RNAfold webserver was used under standard conditions [32].

## 3. Results

### 3.1. Morphological Description

*Pegethrix qiandaoensis* K. Gao, Y. Cheng et R. Li sp. nov. (Figure 1).

The colony appears bright blue-green or olive-green. The filaments are long and vary in width between young and mature trichomes. Morphological images confirmed that no obvious nodule cells or knots were seen in the filaments of the studied strains. Occasionally, the filaments may have single or double false branches, and, in rare cases, the strains show several separate trichomes (Figure 1c). The width of the trichomes is typically between 2.3 and 4.0 μm. The sheath is firm and usually attached to the trichome, although it may occasionally widen or exhibit irregular and stratified patterns (Figure 1h,i). The sheath is absent in immature filaments or hormogonia. The trichomes themselves are untapered, with no or slight constriction at distinct cross walls (Figure 1e,i), and do not form nodules. This trait sets it apart from other *Pegethrix* species. The hormogonia are short (Figure 1g). Cells are usually isodiametric, often shorter than wide, exclusively in meristematic zones and slightly longer than wide in young trichomes. The length of the cells ranges from (1.3)–1.7 to 2.7 μm. End cells were rounded. The thylakoids are parietally arranged (four or five per cell) (Figure 2).

Reference strains: WZU0719 and WZU0723 (The Algae Culture Collection of the College of Life and Environmental Science, Wenzhou University, China).

Type locality: Isolated from the water surface of Qiandao Lake in Zhejiang Province, China (118°34′ E, 29°22′ N).

Holotype designated here: Specimen No. WZUH-ZLYN202312, containing dry material of the strain *Pegethrix qiandaoensis* WZU 0719, has been preserved in the Herbarium of Wenzhou University (WZUH) located in Wenzhou, Zhejiang Province, China.

Etymology: *qiandaoensis* refers to the Qiandao lake where the strain was isolated, transliterated into Latin.

Habitat: Free-living in water surface.

### 3.2. Molecular and Phylogenetic Analyses

Based on the 16S rRNA gene sequences, a distance matrix revealed that the *Pegethrix qiandaoensis* strains had a resemblance of 96.73–97.36% to other *Pegethrix* species (Table 1). A total of 79 representative taxa sequences were examined in the phylogenetic analysis to evaluate the placement of the *Pegethrix* clade among the cyanobacteria (Figure 3). The results of phylogenetic analyses, including MP, ML, and BI analyses, exhibited consistent

tree topologies. The phylogeny constructed using the 16S rRNA gene clearly demonstrates that our strain resided within the genus *Pegethrix* (MP and ML bootstrap percentages of 100% and 100% (BP) and posterior probability (PP) 1.00). The *Pegethrix qiandaoensis* clade occupies an independent phylogenetic position in *Pegethrix*. For a separate ITS phylogenetic tree (Figure 4), the *Pegethrix qiandaoensis* clade can also occupy an independent phylogenetic site within the genus. The differentiation of it has been widely used as molecular evidence of species differentiation at the genus level.

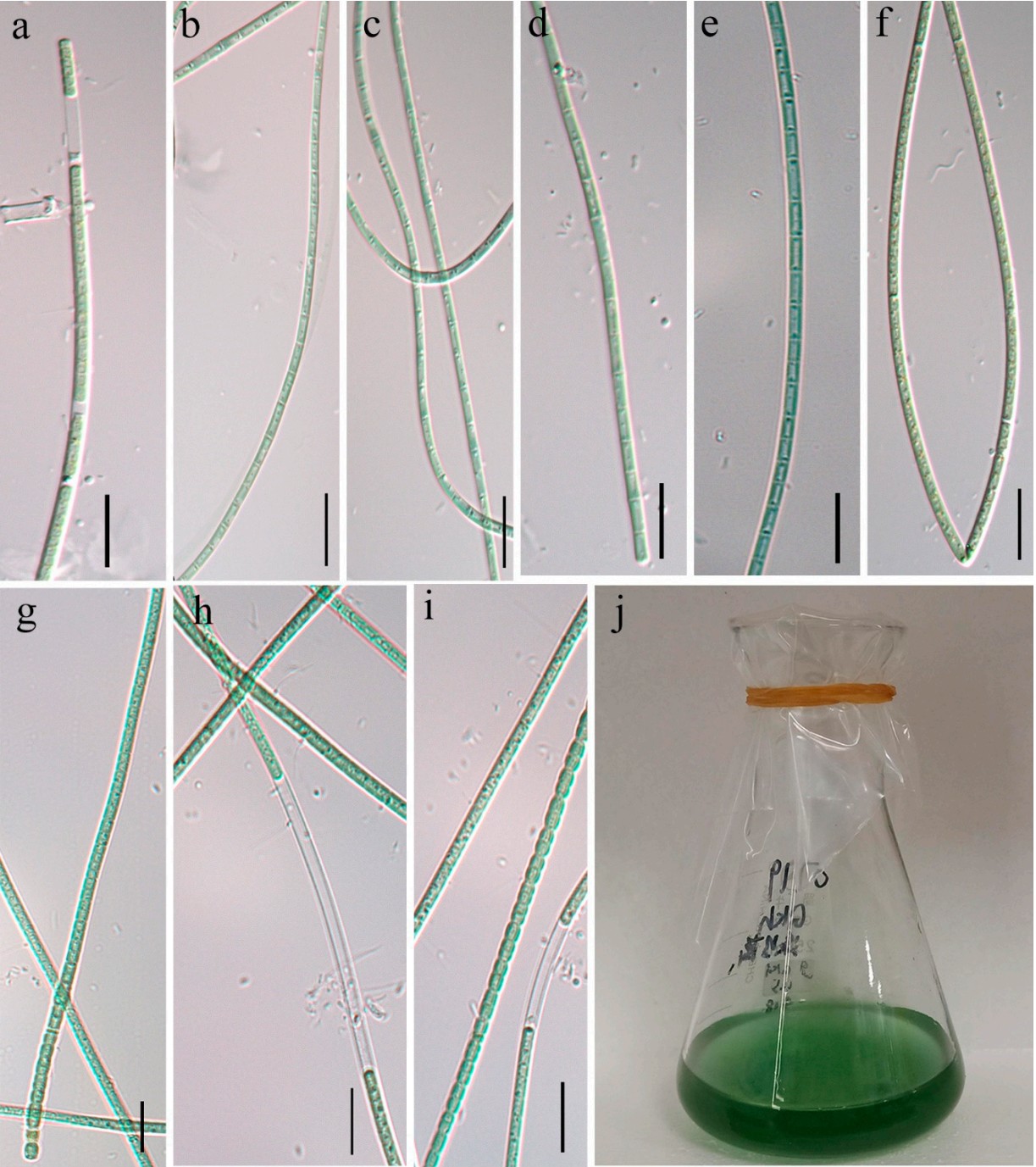

**Figure 1.** Light microscopy of *Pegethrix qiandaoensis* strains. A single trichome with a sheath (**a**,**h**,**i**). Morphological photos confirm that the strains that were studied have no obvious nodules cells and no knots have been observed in the filaments (**a–i**). Observed obvious cell wall structure (**e**,**g**,**i**). Macro-culture status of *Pegethrix qiandaoensis* strains (**j**). Scale bars: 10 μm.

**Table 1.** Comparison of the 16S rRNA gene sequence similarity between *Pegethrix qiandaoensis* and closely related species. Similarity = [1 − (p-distance)] × 100.

| Strain | 1 | 2 | 3 | 4 | 5 | 6 | 7 | 8 | 9 | 10 |
|---|---|---|---|---|---|---|---|---|---|---|
| 1. PP084669 *Pegethrix qiandaoensis* WZU 0719 | | | | | | | | | | |
| 2. OR815386 *Pegethrix qiandaoensis* WZU 0723 | 100.00% | | | | | | | | | |
| 3. OP410738.1 *Pegethrix sichuanica* CCNU0013 | 96.54% | 96.54% | | | | | | | | |
| 4. MT176728.2 *Pegethrix atlantica* BACA0077 | 96.75% | 96.75% | 98.64% | | | | | | | |
| 5. KY078768.1 *Pegethrix bostrychoides* GSE-PSE-MK47-15B | 97.38% | 97.38% | 98.01% | 98.22% | | | | | | |
| 6. KY078764.1 *Pegethrix convoluta* GSE-PSE-MK22-07D | 96.75% | 96.75% | 98.53% | 98.32% | 98.43% | | | | | |
| 7. KY078767.1 *Pegethrix indistincta* GSE-TBC-07GB | 96.86% | 96.86% | 98.64% | 98.43% | 98.53% | 99.90% | | | | |
| 8. HM018688.1 *Pegethrix olivacea* GSE-PSE-MK46-15A | 97.17% | 97.17% | 98.22% | 98.01% | 98.74% | 98.64% | 98.74% | | | |
| 9. NR172705.1 *Trichotorquatus maritimus* SMER-A | 91.51% | 91.51% | 90.46% | 90.46% | 90.04% | 90.67% | 90.78% | 90.46% | | |
| 10. HM018690.1 *Drouetiella lurida* Lukesova 1986/6 | 93.50% | 93.50% | 93.61% | 93.19% | 93.19% | 93.40% | 93.50% | 92.98% | 91.51% | |
| 11. HQ917692.1 *Oculatella subterranea* SP1402/Zammit 2007/5 | 92.14% | 92.14% | 92.45% | 92.24% | 92.24% | 92.98% | 92.87% | 92.35% | 91.72% | 92.24% |

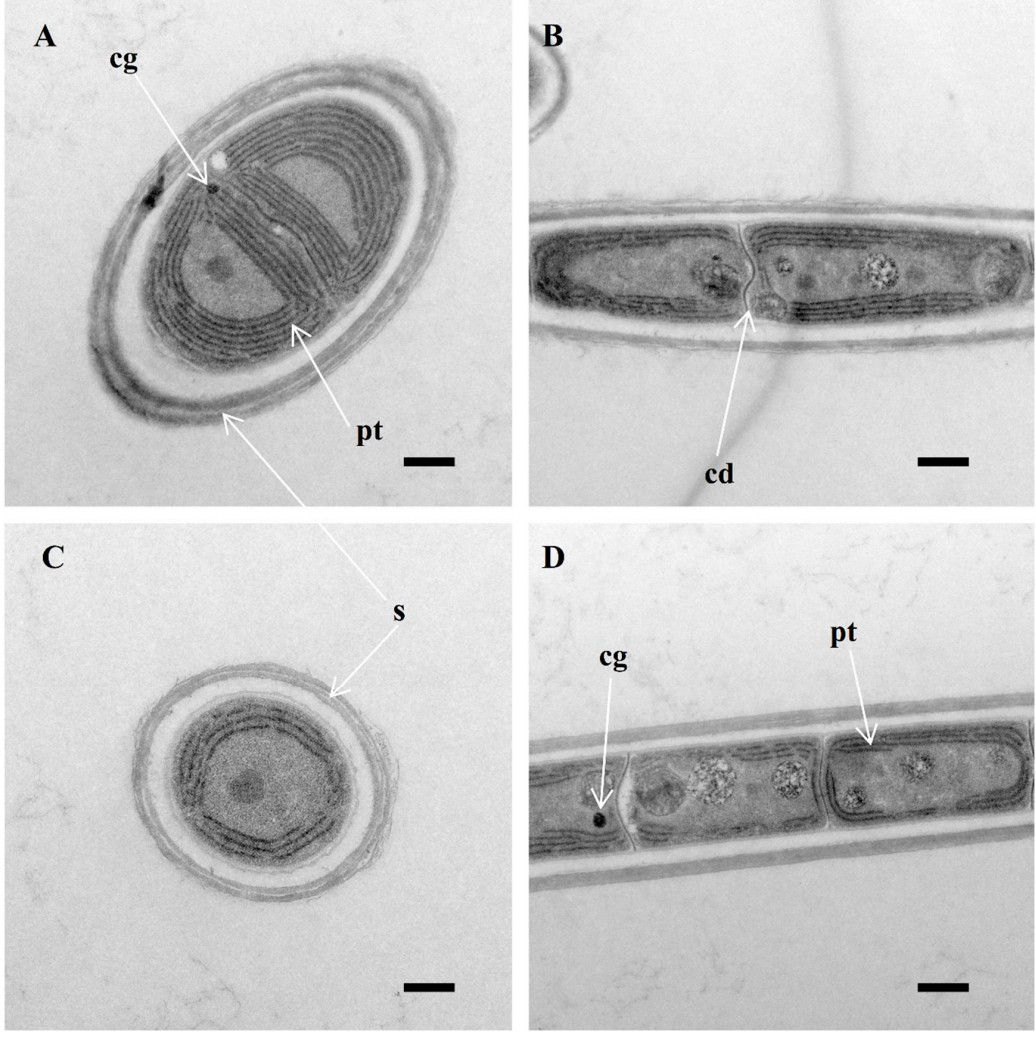

**Figure 2.** Longitudinal section of vegetative cell (TEM), showing sheath (s) and parietal thylakoids (pt) (**A,C,D**), cyanophycin granules (cg) (**A,D**) and cell division—formation of new cell wall (cd) (**B**), and a lack of cell constriction. Scale bar: 0.5 µm.

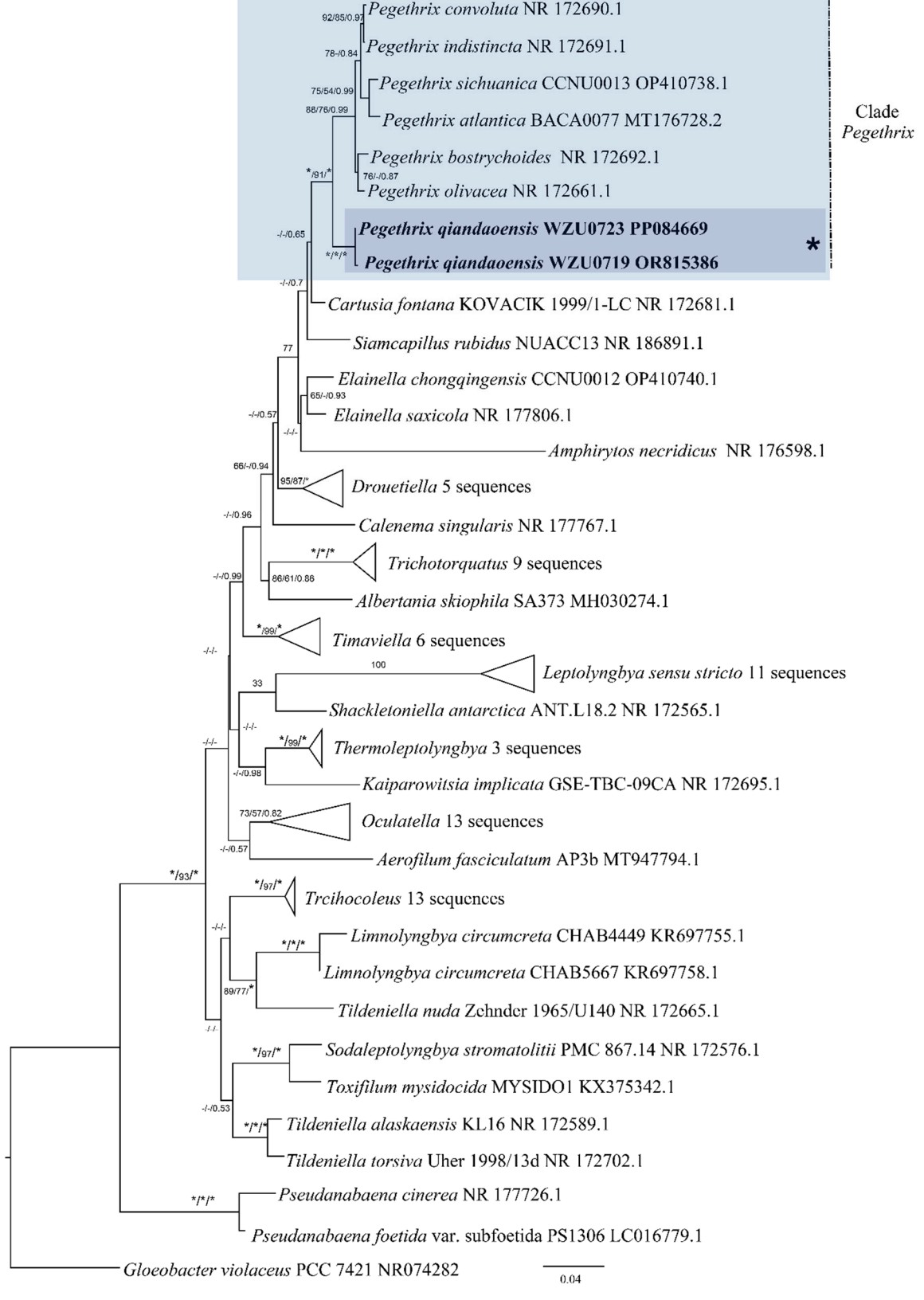

**Figure 3.** Maximum-likelihood (ML) phylogenetic tree of *Pegethrix qiandaoensis* strains based on 16S rRNA gene sequences. For MP/ML methods and Bayesian posterior probabilities, bootstrap values above 50% are displayed in the BI tree. * Indicates a bootstrap value of 100 and a posterior probability of 1.00 for MP, ML, and BI. The novel filamentous strains of this study are indicated in bold.

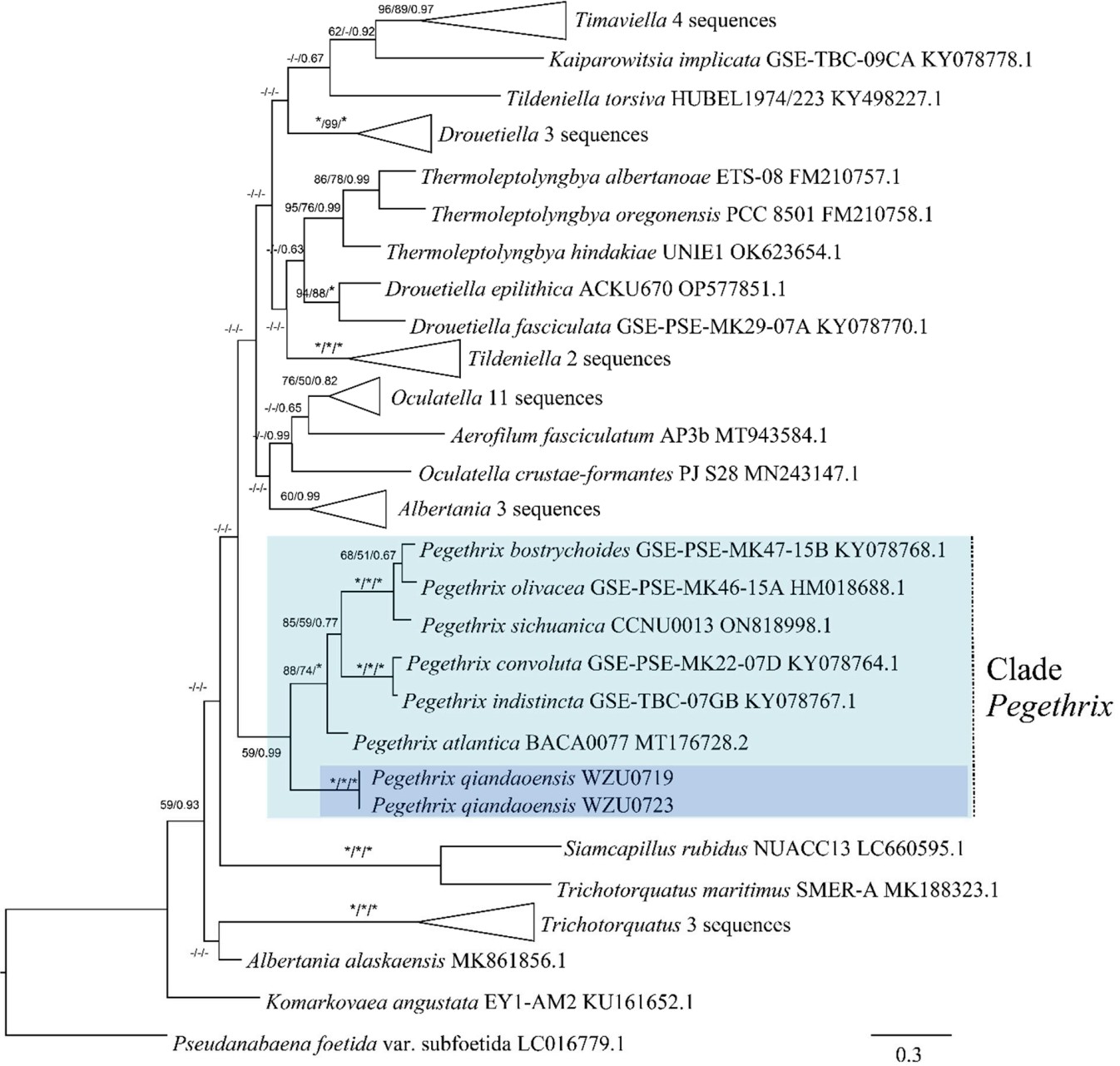

**Figure 4.** Bayesian inference (BI) phylogenetic tree of *Pegethrix qiandaoensis* strains based on ITS rRNA gene sequences. For MP/ML methods and Bayesian posterior probabilities, bootstrap values above 50% are displayed in the BI tree. * Indicates a bootstrap value of 100 and a posterior probability of 1.00 for MP, ML, and BI. The novel filamentous strains of this study are indicated in bold.

### 3.3. Molecular Analyses on the ITS between 16S–23S rRNA Genes

The total length of the Pegethrix qiandanensis strains ITS clones obtained in this study was 649 bp (Table 2), and all sequences contained tRNA Ile and tRNA Ala. The 16S–23S rRNA ITS secondary structures show some similarity among species, even for the D1–D1′ helix (Figure 5) and Box-B helix (Figure 6), with the highest variability found in the V3 region (Figure 7). *Pegethrix qiandaoensis* has the largest top ring structure (5′-AACGAAUAA-3′). *P. atlantica* and *P. bostrychoides* exhibit similar basolateral bulges on the D1–D1′ helix (Figure 5), a feature unique to these species, although there are some differences in the central inner loop. The terminal hairpin of *Pegethrix* contains four-nucleotide residues (5′-GGAA-3′) found in *P. convoluta*, *P. atlantica*, and *P. bostrychoides*, and the same nucleotide

residue is found in *P. convoluta* an *P. indistincta* (5′-GAGA-3′). *P. sichuanica* has a unique nucleotide residue (5′-GGAA-3′).

**Table 2.** Analysis of the length (number of nucleotides) of the 16S-23S ITS region of *Pegethrix* strains.

| Strain | Complete ITS | D1-D1′ Helix | D2 | tRNA Ile | V2 Spacer | tRNA Ala | Box-B Helix | Box-A Helix | D4 | V3 Helix |
|---|---|---|---|---|---|---|---|---|---|---|
| 1. *Pegethrix qiandaoensis* WZU 0719 | 649 | 87 | 12 | 74 | 60 | 73 | 50 | 12 | 7 | 126 |
| 2. *Pegethrix qiandaoensis* WZU 0723 | 649 | 87 | 12 | 74 | 60 | 73 | 50 | 12 | 7 | 126 |
| 3. KY078768.1 *Pegethrix bostrychoides* GSE-PSE-MK47-15B | 614 | 87 | 12 | 74 | 36 | 73 | 36 | 12 | 7 | 94 |
| 4. KY078764.1 *Pegethrix convoluta* GSE-PSE-MK22-07D | 627 | 91 | 12 | 74 | 14 | 73 | 36 | 12 | 7 | 108 |
| 5. KY078767.1 *Pegethrix indistincta* GSE-TBC-07GB | 627 | 91 | 12 | 74 | 14 | 73 | 36 | 12 | 7 | 108 |
| 6. HM018688.1 *Pegethrix olivacea* GSE-PSE-MK46-15A | 614 | 87 | 12 | 74 | 36 | 73 | 36 | 12 | 7 | 94 |
| 7. OP410738.1 *Pegethrix sichuanica* CCNU0013 | 559 | 87 | 12 | 74 | 15 | 73 | 38 | 12 | 7 | 95 |
| 8. MT176728.2 *Pegethrix atlantica* BACA0077 | 630 | 91 | 12 | 74 | 29 | 73 | 33 | 12 | 7 | 96 |

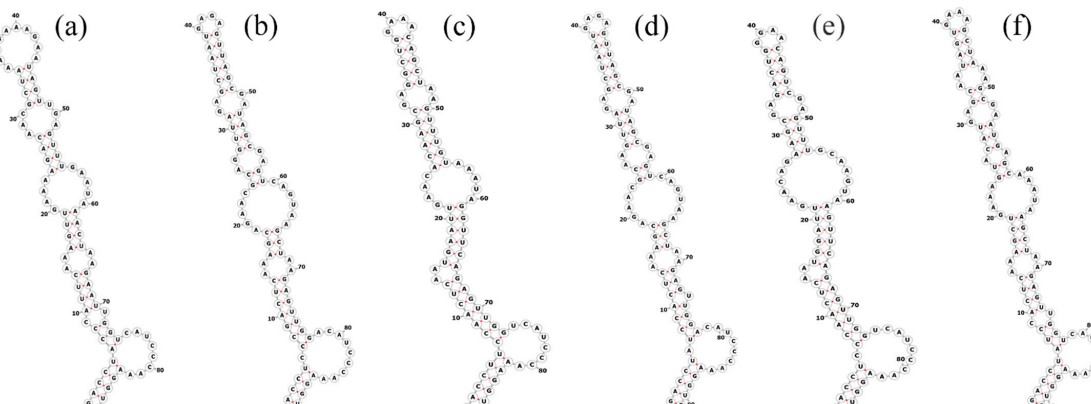

**Figure 5.** Secondary structures of the D1–D1′ helix in *Pegethrix* species. (**a–f**) D1–D1′ helices: (**a**) *Pegethrix qiandaoensis* WZU0719. (**b**) *Pegethrix convoluta*. (**c**) *Pegethrix bostrychoides*. (**d**) *Pegethrix indistincta*. (**e**) *Pegethrix sichuanica*. (**f**) *Pegethrix atlantica*.

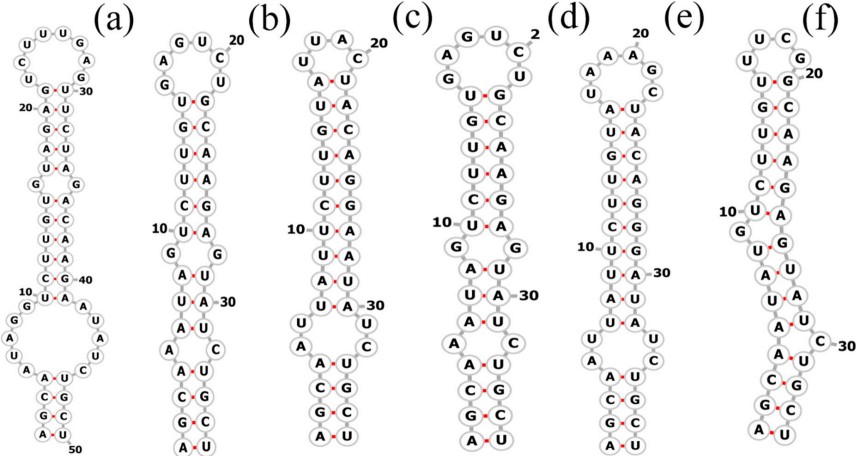

**Figure 6.** Secondary structures of the Box–B helix in *Pegethrix* species. (**a–f**) D1–D1′ helices: (**a**) *Pegethrix qiandaoensis* WZU0719. (**b**) *Pegethrix convoluta*. (**c**) *Pegethrix bostrychoides*. (**d**) *Pegethrix indistincta*. (**e**) *Pegethrix sichuanica*. (**f**) *Pegethrix atlantica*.

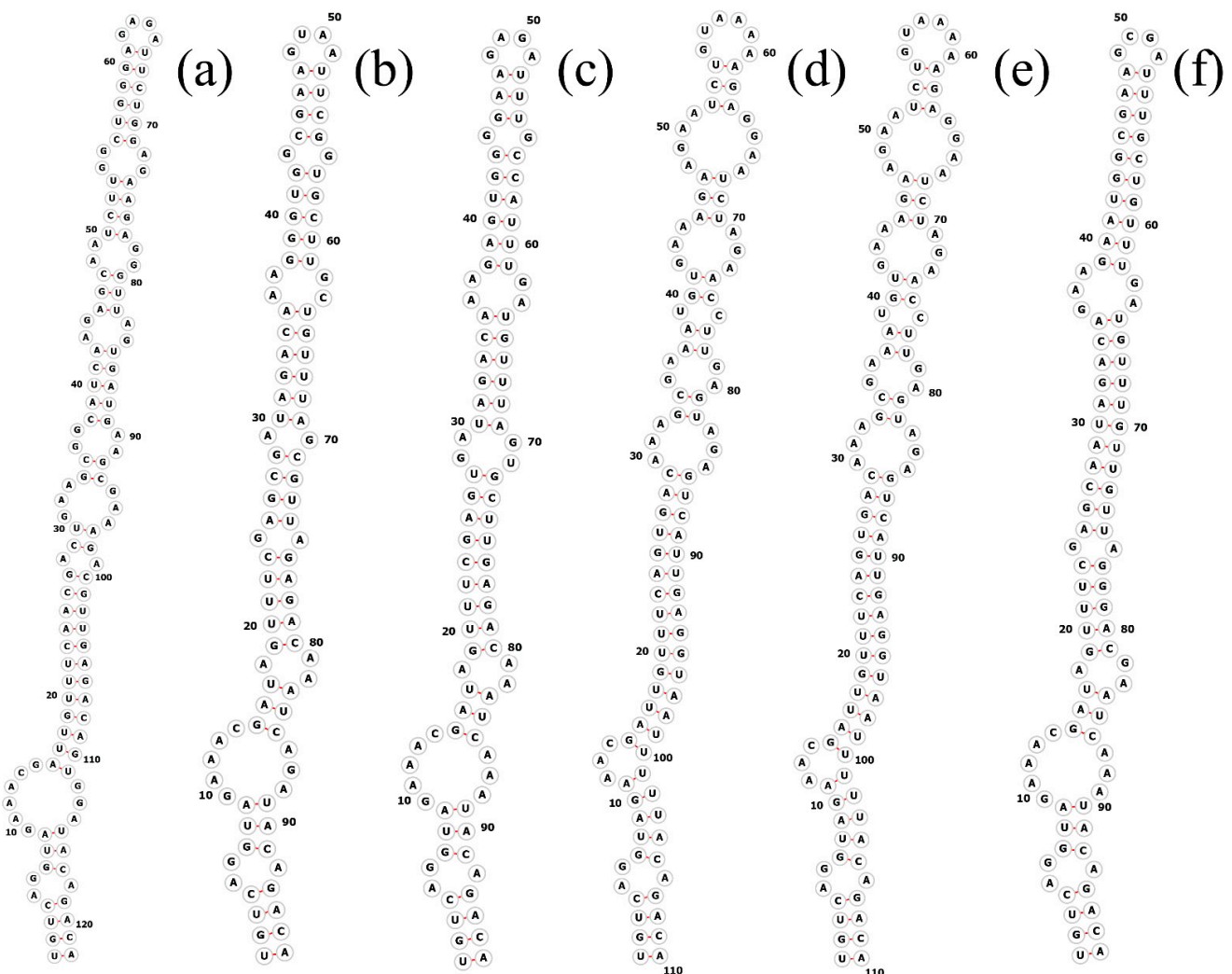

**Figure 7.** Secondary structures of the V3 helix in Pegethrix species. (**a**–**f**) D1–D1′ helices: (**a**) *Pegethrix qiandaoensis* WZU0719. (**b**) *Pegethrix convoluta*. (**c**) *Pegethrix bostrychoides*. (**d**) *Pegethrix indistincta*. (**e**) *Pegethrix sichuanica*. (**f**) *Pegethrix atlantica*.

## 4. Discussion

To establish the cyanobacterial taxonomic system consisting of monophyletic groups at all ranks is becoming an ideal aim for studying the true diversity of cyanobacteria [33,34]. Further, with the usage of the polyphasic approach, a large number of cyanobacterial new genera and species has been described, which subsequently led to substantial taxonomic revisions, especially at the order and family levels [4,14,16]. In particular, thin filamentous cyanobacteria, previously only belonging to Oscillatoriales, have been divided into several orders in the updated taxonomic system [16], including Nodosilineales, Oculatellales and Leptolyngbyales. In this study, we present a scientific investigation that focuses on examining and categorizing two strains of thin filamentous cyanobacteria through the polyphasic approach. Based on a comprehensive analysis, we have identified them within the genus *Pegethrix*, Oculatellaceae, and further established them as a novel species. This conclusion is reached by comprehensively considering the sequence threshold of the 16S rRNA gene, the phylogeny based on the 16S rRNA gene, and the secondary structure and morphological characteristics of ITS [35,36]. The results of this study also revealed that species with similar phylogenetic positions can exhibit significant differences in habitat performance.

Since its establishment in 2018, the genus *Pegethrix* has been shown to exhibit remarkable similarities among species. Because of the uncomplicated morphological traits within this genus, some overlapping characteristics exist among the species. Nonetheless, specific morphological attributes provide valuable diagnostic clues for identifying *Pegethrix qiandaoensis* in this study. These attributes comprise the development of elongated, loosely coiled filaments under typical lighting conditions, as well as the presence of a stratified and expanded sheath. Moreover, the filaments in *P. qiandaoensis* hardly knot to form nodules, revealing the obvious difference from other *Pegethrix* species. Phylogenetically, the close clade formed by the strains of *P. qiandaoensis* and the other six *Pegethrix* species based on the 16S rRNA gene, with high support values, represents the strong monophyletic group for the genus *Pegethrix* (Figure 4). Moreover, 96.73–97.36% of the 16S rRNA similarities among *P. qiandaoensis* and the other *Pegethrix* species, lower than all the proposed cut-off values of bacterial species separation [37,38], strongly support the establishment of *Pegethrix qiandaoensis* sp. nov. From the initial establishment of the genus *Pegethrix* described in Mai et al. (2018) [17], the sequence identities of the 16S rRNA gene sequences among the six species of the genus, ranging from 98.6 to 99.9%, did not provide evidence of species separation in this genus using only these sequence identities. However, the addition of *Pegethrix qiandaoensis* sp. nov. in this study can help considerably to molecularly provide a high validity of the genus *Pegethrix*. The 16S–23S ITS region plays a crucial role in distinguishing between different species [8,39]. The results from ITS analyses indicate that each species has distinct secondary structures in the 16S-23S ITS region, specifically in the D1-D1' and Box-B helices. These structural differences, including the significantly different Box-B helix observed in *P. qiandaoensis* compared to the other *Pegethrix* species (Figures 6 and 7), provide further evidence of significant genetic divergence among various lineages [8,40,41]. This observation holds true for all the genera examined and discussed in this research, highlighting the consistent reliability and sensitivity of the ITS gene in distinguishing between genera.

Last, from the ecological viewpoint, the occurrence of this species in a freshwater ecosystem in China further distinguished it from previously documented species, which primarily inhabit soil, stones, rocks, and thermal springs. This study also revealed that species with a similar phylogenetic development relationship can exhibit significant differences in habitat performance. Apparently, *P. qiandaoensis* in this study was shown to stand as the most distinctive member in the genus *Pegethrix*.

In conclusion, the present study provided a typical case with the focus on diversity of a cyanobacterial genus with thin filaments, based on the polyphasic approach. In recent years, a rapidly growing number of new taxa of the thin filamentous cyanobacteria have been intensively reported and progress in the taxonomy of such cyanobacterial groups has led to considerable revisions in cyanobacterial taxonomy. Although only one species new to science was recognized in this study from a genus of the thin filamentous cyanobacteria, it is clear that more species, genera, or high categories in these cyanobacterial forms remain to be named and defined. With enhancement of the polyphasic culture-dependent approach, more results on the diversity of thin filamentous cyanobacteria from less explored regions in China will be highly expected.

**Author Contributions:** Conceptualization, K.G. and Y.C.; methodology, Y.C.; software, Y.C. and P.X.; formal analysis, Y.C. and R.G.; investigation, K.G. and Y.C.; resources, Z.W. and H.Z.; data curation, Y.C.; writing—original draft preparation, K.G. and Y.C.; writing—review and editing, F.C. and R.L.; visualization, F.C.; funding acquisition, R.L. All authors have read and agreed to the published version of the manuscript.

**Funding:** This study is funded by the Zhejiang Provincial Natural Science Foundation of China (LD21C030001).

**Institutional Review Board Statement:** Not applicable.

**Informed Consent Statement:** Not applicable.

**Data Availability Statement:** The 16S rRNA gene sequences are available in the GenBank database.

**Acknowledgments:** The first author thanks Huimin Fang and Yuping Fan for physical assistance in the process of collecting samples, and thanks Zhenfei Xing for technical assistance with the TEM images.

**Conflicts of Interest:** The authors declare no conflicts of interest.

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
