# Peer review of "Revealing the Diversity of Thin Filamentous Cyanobacteria, with the Discovery of a Novel Species, Pegethrix qiandaoensis sp. nov. (Oculatellaceae, Oculatellales), in a Freshwater Lake in China"

_diversity, doi:10.3390/d16030161_

Round 1
Reviewer 1 Report
Comments and Suggestions for Authors
The peer-reviewed manuscript of Gao et al. describes a new species from a recently established genus Pegethrix Mai, J.R. Johansen et Bohunická, 2018. Two Leptolyngbya-like cyanobacterial strains were isolated from the surface of freshwater Qiandao Lake in China and studied using the polyphasic approach. Based on thorough morphological and phylogenetic analyzes, authors proposed new representative of this genus with geographically referenced species epithet: Pegethrix qiandaoensis sp. nov. The study not only expands the species diversity of Pegethrix, but also its ecological spectrum: from terrestrial to terrestrial-aquatic. The MS completely fit the Diversity scope; it is based on original data obtained as a result of a multifaceted study of the isolated strains. The MS makes a good impression. The material is presented compactly and at the same time all stages of the work are given in details. The methods are adequate, the results look reliable, their interpretation is well justified. The bibliography contains only the necessary key sources. The tables and figures are clear and mostly well labeled (except of Figures 1, 5-7: see comments in MS). The text also contains typos and other omissions; they are noted in the manuscript. After eliminating the shortcomings noted above, I recommend to accept the paper of Gao et al. “Revealing the diversity of thin filamentous cyanobacteria, with discovery of a novel species as Pegethrix qiandaoensis sp. nov. (Oculatellaceae, Oculatellales) from a freshwater lake in China” for publication.

Author Response
Response to Reviewer 1 Comments
Point 1: Annotation errors in Figure 1.
Response: The annotation in Figure 1 has been revised according to the review comments (Line 200-203).
Point 2: Annotation errors in Figure 5.
Response: The species name errors and format errors in the annotation in Figure 5 have been modified according to the review comments (Line 252-253).
Point 3: Annotation errors in Figure 6.
Response: The species name errors and format errors in the annotation in Figure 6 have been modified according to the review comments (Line 255-256).
Point 4: Annotation error in Figure 7.
Response: The species name errors and format errors in the annotation in Figure 7 have been modified according to the review comments (Line 258-259).
Point 5: Spelling errors and species names in the abstract are not italicized.
Response: The spelling errors and species names without italics in the abstract have been corrected (Line 21-29).
Point 6: The marked keyword needs to be replaced.
Response: “Pegethrix” has been replaced by “new species” in the keywords.
Point 7: There are formatting errors and inappropriate wording errors in the introduction of the manuscript.
Response: The format errors and spelling errors in the introduction part of the manuscript have been modified according to the requirements of review (Line 60, 67, 72). Replace the word “freshwater” with “lake” (Line 64).
Point 8: There are incorrect words usage, expression errors and formatting errors in the materials and methods part of the manuscript.
Response: The corresponding words replacement is made where the words are improperly used (Line 77, 80, 84, 86, 96, 141). The text expression has been modified according to the requirements of the review (Line 94). All marked format errors have been modified (Line 100, 114, 134).
Point 9: There are incorrect expressions and some formatting errors in the result part of the manuscript.
Response: The morphological description part of the results of the manuscript has been modified in accordance with the requirements of the review (Line 176-182, 187). All the marked format errors in the result part of the manuscript have been modified (Line 190, 212).
Point 10: There are singular and plural expression errors in the annotations of Figure 3 and Figure 4 in the manuscript.
Response: The singular and plural expression errors in the annotations of Figure 3 and Figure 4 in the manuscript have been modified (Line 228, 234).
Point 11: There are word errors and format errors in the discussion part of the manuscript.
Response: The expression errors and format errors in the discussion part of the manuscript have been modified (Line 267, 286, 305, 312, 318).
Point 12: There are some format errors in the reference part of the manuscript.
Response: These errors have been corrected (Line 350, 354, 364, 389, 427).
Thank you again for your excellent review work.

Reviewer 2 Report
Comments and Suggestions for Authors
In the reviewed paper very interesting and important for a taxonomy of filamentous cyanobacteria results are presented. The authors performed the study of two strains from a freshwater Lake in China, using a polyphasic approach (light microscopy, molecular-genetic and TEM analysis). The conclusions of the MS are supported by results. The reference list contains the basic publications on the studied topic.
But the quality of the results presentation should be improved. Besides, the MS contains many errors.
Suggestions for the authors:
Line 21: Correct “incor-porating” to “incorporating”.
Line 22: Correct “pro-vided” to “provided”.
Line 22, 25 and further: Latin names should be italicized.
Lines 137-147 and further: When describing methods, please avoid third-person descriptions (we performed, we discarded).
Figure 1: The quality of the figure should be improved. For this purpose, it is better to use Adobe Photoshop software. The distances between the photos should be the same, the scale on all photos should be the same thickness. The photos labelled as A-J, but in the legend it written as a-j. Where are the arrows, which were mentioned in the legend? I think, that it is better to exclude the last picture J or replaced it with the flask with the nice label and cap.
Lines 177-183: Correct ((Figure 1, a, b) to (Figure 1, a, b).
Figure 2: Add information about pictures A-D to the figure legend.
Figure 3,4: Improve resolution of phylogenetic trees.
Lines 238-241: Correct ((Figure 5) to (Figure 5).
Line 292: Correct “Pegethrix qiandaoensis sp. nov..” to “Pegethrix qiandaoensis sp. nov.”
Reference: Please, correct reference list according to journal requirement. Remove the dots after the journals names.
Comments on the Quality of English LanguageModerate editing of English language required
Author Response
Response to Reviewer 2 Comments
Point 1: Line 21: Correct “incor-porating” to “incorporating”.
Response: This error has been corrected (Line 21).
Point 2: Line 22: Correct “pro-vided” to “provided”.
Response: This error has been corrected (Line 21).
Point 3: Line 22, 25 and further: Latin names should be italicized.
Response: This error has been corrected (Line 25, 26).
Point 4: Lines 137-147 and further: When describing methods, please avoid third-person descriptions (we performed, we discarded).
Response: The expression of these parts has been modified according to the revised opinions, please see the revised text (Line 136-147).
Point 5: Figure 1: The quality of the figure should be improved. For this purpose, it is better to use Adobe Photoshop software. The distances between the photos should be the same, the scale on all photos should be the same thickness. The photos labelled as A-J, but in the legend, it written as a-j. Where are the arrows, which were mentioned in the legend? I think that it is better to exclude the last picture J or replaced it with the flask with the nice label and cap.
Response: Thank you for pointing out the problem with the picture, and the picture notes have been replaced according to your suggestions.
Point 6: Lines 177-183: Correct ((Figure 1, a, b) to (Figure 1, a, b).
Response: This error has been corrected (Line 176-182).
Point 7: Figure 2: Add information about pictures A-D to the figure legend.
Response: Some information has been added to the picture annotations according to your comments (Line 220-221).
Point 8: Figure 3,4: Improve resolution of phylogenetic trees.
Response: In the corresponding position, the higher resolution picture has been replaced.
Point 9: Lines 238-241: Correct ((Figure 5) to (Figure 5).
Response: This error has been corrected (Line 242-245).
Point 10: Line 292: Correct “Pegethrix qiandaoensis sp. nov..” to “Pegethrix qiandaoensis sp. nov.”
Response: This error has been corrected (Line 293).
Point 11: Reference: Please, correct reference list according to journal requirement. Remove the dots after the journal’s names.
Response: Revision for the references list has been done accordingly.
Thank you again for your excellent review work.
